# Australian General Practitioners’ Current Knowledge, Understanding, and Feelings Regarding Breast Density Information and Notification: A Cross-Sectional Study

**DOI:** 10.3390/ijerph19159029

**Published:** 2022-07-25

**Authors:** Hankiz Dolan, Kirsten McCaffery, Nehmat Houssami, Meagan Brennan, Melanie Dorrington, Erin Cvejic, Jolyn Hersch, Angela Verde, Lisa Vaccaro, Brooke Nickel

**Affiliations:** 1Wiser Healthcare, Sydney School of Public Health, Faculty of Medicine and Health, The University of Sydney, Sydney 2006, Australia; hankiz.dolan@gmail.com (H.D.); kirsten.mccaffery@sydney.edu.au (K.M.); nehmat.houssami@sydney.edu.au (N.H.); erin.cvejic@sydney.edu.au (E.C.); jolyn.hersch@sydney.edu.au (J.H.); 2Sydney Health Literacy Lab, Sydney School of Public Health, Faculty of Medicine and Health, The University of Sydney, Sydney 2006, Australia; 3The Daffodil Centre, The University of Sydney, A Joint Venture with Cancer Council NSW, Sydney 2006, Australia; 4School of Medicine Sydney, University of Notre Dame Australia, Sydney 2007, Australia; meagan.brennan@sydney.edu.au; 5Westmead Breast Cancer Institute, Westmead Hospital, Sydney 2145, Australia; 6Bungendore Medical Centre, Bungendore 2621, Australia; drmelaniedorrington@gmail.com; 7Breast Cancer Network Australia, Melbourne 3124, Australia; angela@verde.id.au; 8Health Consumers New South Wales, Sydney 2000, Australia; lisavac@bigpond.com; 9Discipline of Behavioural and Social Sciences in Health, Sydney School of Health Sciences, Faculty of Medicine and Health, The University of Sydney, Sydney 2006, Australia

**Keywords:** breast density, general practice, mammography, knowledge, notification

## Abstract

Background: There is a lack of evidence around Australian general practitioners’ (GPs) views of issues surrounding breast density. The current study aimed to quantitatively assess GPs’ current knowledge, understanding, and feelings around breast density information and notification. Methods: This study involved a cross-sectional survey using an online platform to collect quantitative data from Australian GPs. Survey data were analysed with descriptive statistics. Results: A total 60 responses from GPs were analysed. Most (*n* = 58; 97%) had heard or read about breast density and nearly 90% (*n* = 52; 87%) have had discussions about breast density with patients. Three-quarters (*n* = 45; 75%) were supportive of making breast density notification mandatory for patients with dense tissue and a similar proportion (*n* = 45/58; 78%) felt they need or want more education on breast density. Conclusions: There is strong support for notifying patients of breast density, and interest in further education and training among the surveyed GPs. As GPs play a central role in cancer prevention and control, their involvement in discussions related to breast density notification, evaluation and appraisal of evidence, development of communication strategies, and participation in ongoing research on the topic will be indispensable.

## 1. Introduction

Breast density is an independent risk factor for developing breast cancer [1]. A density level (entirely fatty, scattered areas of fibroglandual density, heterogeneously dense, or extremely dense) is usually determined by radiologists using mammography [2]. As mammography uses X-ray, the more dense a tissue is, the whiter it appears, hence denser breast tissue can ‘white-out’ or obscure a cancerous lesion, which also appear white. Mammography has been associated with lower sensitivity in detecting breast cancer in individuals with denser breast tissue, and an increased risk of breast cancer diagnosis between routine screening intervals [3,4].

Breast density is a relatively non-modifiable risk factor. Currently, there are no uniformly recommended treatment or prevention options related to breast density [5]. Supplemental screening, such as offering ultrasound or MRI in addition to mammography, may be offered to women with dense breasts to increase screening sensitivity. However, the evidence surrounding the benefits and harms of supplemental screening remains unclear. Supplemental screening may increase cancer detection rates; however, it can increase the rates of false-positive results, potentially leading to overdiagnosis and overtreatment [6]. Furthermore, there is still uncertainty around whether supplemental screening improves long-term health outcomes, including the breast cancer mortality rate.

Recently, there has been discussion about whether population-based screening program participants should be notified of their breast density. Much of this debate is around the potential benefits and harms of notification, including health system effects, clinical outcomes, aspects of medical decision-making, and individual psychological and wellbeing impact. The United States (US) Congress recently passed legislation requiring mammography facilities to inform individuals, and their physicians, of their breast density level [7]. To date, no other country has legislated mandated notification of breast density.

Currently, breast density is neither measured nor reported in the Australian publicly funded breast screening program (BreastScreen Australia), with the exception of BreastScreen WA. Having undertaken recent evidence reviews, BreastScreen Australia’s Standing Committee on Screening recommended that breast density should not be routinely recorded [8,9]. It also recommends that, given the uncertainties in the measurement and management of breast density, patients with dense breasts not be offered supplemental screening (BreastScreen Australia 2016, 2020). However, BreastScreen Australia also noted in the statement that it will continue to engage key stakeholders in developing an evidence base and potentially piloting notification [8,9]. There have been consumer groups advocating for breast density notification, and some women have already been being notified through private screening services.

In Australia, quantitative evidence on the impact of breast density information and notification for primary care practitioners is lacking, despite heightened debate around breast density notification. General practitioners (GPs) are at the forefront of primary and secondary prevention of breast cancer in Australia and are likely to become a first point-of-contact for patients who are informed of, or are concerned about, their breast density. Therefore, the current study aimed to quantitatively assess Australian GP’s current knowledge, experience/practice, and attitudes around breast density information and notification.

## 2. Materials and Methods

### 2.1. Design

This is a cross-sectional online survey study. Ethical approval was granted by the University of Sydney Human Research Ethics Committee (2021/001).

### 2.2. Sample and Recruitment

GPs were recruited through several avenues, including (1) an email, with the participant information statement (PIS) attached, to the Royal Australian College of General Practitioners Breast Medicine Special Interest Group mailing list; (2) an email, with PIS attached, to a list of GPs who had consented to be contacted for future studies after participating in previous University of Sydney research; and (3) a brief study advert on the ‘GPs DownUnder’ and ‘GP Mums (AUS/NZ)’ Facebook group pages, which together have more than 10,000 active GP members. We also contacted all 31 primary health networks across Australia and received permission or confirmation from nine of them to advertise either through their website or newsletters. All avenues included study investigators’ details and a link to the survey landing page on the Qualtrics online platform [10]. The survey landing page included brief information about the study along with the PIS and participant consent form (PCF) for downloading and viewing. Each participant was offered a chance to enter a prize draw to win one of five gift vouchers ($A100 each).

### 2.3. Data Collection

The questionnaire consisted of measurement scales and items adapted from previously published literature [11,12,13,14] and self-developed items based on our previous qualitative study findings (see Appendix A: Survey for primary care clinicians (GPs)). The questionnaire was divided into five sections: (1) GP characteristics; (2) knowledge of breast density, where those with self-identified knowledge of breast density were asked further questions about the source of the knowledge, and regarding breast density measurement, association with cancer risk and age, impact on mammogram reading, and supplemental screening (GPs who had indicated no prior knowledge of breast density skipped to section four); (3) prior experience and practices in discussing breast density with women (the term woman is used to include all people who have experienced oestrogen dependent breast tissue development); (4) attitudes and views towards notification, where GPs were provided with a short synopsis of the current evidence surrounding breast density and the current landscape of notification in Australia and overseas; and (5) the need for future GP education and training, as well as suggestions. The online survey was open for data collection from May to November 2021.

### 2.4. Data Analysis

Data were descriptively analysed using SPSS version 24 [15]. Frequency and relative frequency were used to summarise binary and categorical variables. Continuous variables are reported using means and standard deviations.

## 3. Results

### 3.1. Participant Characteristics

Sixty valid and near-complete (progress ≥ 80%) responses were received. Of the GPs who participated, 45 (75%) worked exclusively in private practices, 40 (69%) practiced in major cities, 54 (90%) were female, half had been working in general practice for less than 10 years, and 22 (37%) had a special interest in women’s health or breast health (Table 1).

### 3.2. Knowledge about Breast Density

Most of the GPs (*n* = 58; 97%) had heard or read about breast density prior to taking the survey. For those who had prior knowledge, ‘reading the mammogram reports’ was the most common source for having heard or read about breast density (*n* = 43; 74%), followed by ‘talking to colleagues or other clinicians’, and ‘reading journal articles or other professional reading materials’ (both *n* = 27; 47%). GPs self-rated their understanding of breast density on a five-point Likert scale and the mean score was 3.3 (range 2–5, standard deviation 0.85), indicating an average level of understanding. The majority of GPs correctly identified how breast density was measured (*n* = 48; 83% for mammogram); one-fifth (*n* = 13; 22%) incorrectly identified physical examination of breasts. More than half of the GPs were aware of increased breast cancer risk associated with density (*n* = 36; 62%); risk of dense breast tissue masking cancer on mammography (*n* = 53; 91%); decrease in breast density with age (*n* = 48; 83%); and the concept of supplemental screening (*n* = 42; 72%) (see Table 2).

### 3.3. Prior Experiences and Practices

Among 58 GPs who reported having prior knowledge of breast density, almost 40% (*n* = 23) reported having discussions with patients about breast density around once a month, and 22.4% (*n* = 13) reported having a discussion once a year (see Table 3). About 10% (*n* = 6) reported having never discussed breast density with a patient for various reasons, including having never seen it reported on mammogram reports. Of those who had discussed breast density with patients (*n* = 52), most were very (*n* = 16; 31%) or somewhat (*n* = 26; 50%) comfortable with answering patients’ questions about breast density. More than half of the GPs with self-identified knowledge regarding breast density (*n* = 28; 54%) reported offering supplemental screening to only certain patients with dense breast tissue, depending on risk factors or on recommendations from the mammography report.

### 3.4. Views towards Notification

Three quarters (*n* = 45; 75%) of all GP respondents were supportive of the notion of making breast density notification mandatory for patients with dense tissue (see Table 4). Most agreed that notifying patients will promote informed decision making (*n* = 44; 76%) and were in favour of notifying via mammogram results (*n* = 46; 79%). The GPs agreed or strongly agreed with the suggestion that counselling patients about breast density is primarily their responsibility as a GP (*n* = 40; 69%), and that patients have a right to know about their breast density (*n* = 53; 91%). Half of the GPs (*n* = 30; 52%) agreed/strongly agreed that notification might cause patients undue anxiety. Almost 70% (*n* = 40) agreed/strongly agreed that a policy of routine and widespread notification of breast density to patients would impact their clinical practice, and a similar proportion (*n* = 39; 67%) felt they were prepared to respond to requests from patients. Most GPs thought that statements to be added to the breast density notification letter sent to patients should include “additional screening may be advisable” (*n* = 44; 76%) and “the patient should discuss results with their primary care physician or the referring physician” (*n* = 48; 83%).

### 3.5. Future Information Provision and Training

More than three-quarters of GPs (*n* = 45/57, 79%) felt they need or want more education on breast density. They felt the most useful options to increase their knowledge would be synopsis of up-to-date scientific evidence (*n* = 40, 70%), informational pamphlets/materials for patients (*n* = 40, 70%), and professional college educational programs (*n* = 37, 65%). Conferences or workshops (*n* = 22, 39%) and interdisciplinary discussions with radiology departments (*n* = 16, 28%) were also noted as useful educational options to alleviate confusion and improve the counselling process.

## 4. Discussion

This cross-sectional survey of Australian GPs shows that, despite a high level of awareness of breast density as an issue, there is still a lack of knowledge around its clinical significance and evidence-based supplemental screening options. Most GPs reported experience of discussing breast density with patients, although with substantial varying frequency. There was strong support for breast density notification, with three in four GPs supporting an Australian mandatory reporting policy. Importantly, over 82% of GPs believed that they were the appropriate clinicians to discuss breast density with patients, with the majority indicating they would need more education on the topic.

Nickel et al. previously conducted a qualitative study into the views and perception of Australian GPs regarding breast density and potential notification [16]. The findings from this study pointed to GPs’ limited knowledge about breast density and mixed views towards notification, especially when clear guidelines in managing breast density are lacking [17]. The findings from the current study, however, indicate a higher level of knowledge and support for notification. This could be because of a high proportion of GPs in the current study having a specific interest in breast or women’s health. When compared with a similar cross-sectional study of primary care clinicians from Massachusetts, United States, where there is mandated notification, a higher proportion of Australian GPs held positive attitudes towards informing patients of their breast density (38% vs. 79.3%), believed it would promote informed-decision making (25% vs. 75.9%), and regarded counselling as their responsibility (43% vs. 68.9%) [13]. Again, these results need to be interpreted in light of our sample characteristics and differences in health systems. Compared with physicians in another survey study from the Mayo Clinic in the United States, GPs in our study had a similar comfort level in discussing breast density with patients (around 80%), yet were less likely to offer supplemental screening based on individual risk assessment (47% vs. 26.9%) [12]. Our findings demonstrate that country-level variation in attitudes and views exists, and Australian GPs in the current study tend to be more positive about breast density notification. However, given the limitations noted below, to draw a more definitive conclusion, more research is needed. Regardless, even among this group of GPs who had self-selected to participate in the current study and had a higher level of awareness or experiences with breast-density-related issues, the interest and need for further education and training is clear. This is in line with our earlier qualitative study findings [16]. This highlights the need for widespread education campaigns, especially if breast density notification is to be introduced through BreastScreen programs.

Despite our efforts to recruit GPs from multiple channels and maximise the sample size, because survey sampling happened during the COVID-19 pandemic, for which GPs were at the frontline, we were only able achieve a small sample size. This has limited our analysis to be only descriptive. We also had a high proportion of GPs (35%) who had a special interest in women’s or breast health. GPs who decided to take part in the study might be more aware of or interested in topics relating to breast density than those who did not. Therefore, although the survey was open to all Australian GPs, the study findings may not be generalisable. Future studies with a larger and more representative sample size are warranted.

## 5. Conclusions

This paper provides a descriptive analysis of data collected from a small group of GPs in Australia regarding their knowledge, attitudes, and views about breast density and its potential notification for patients. The results are promising in terms of awareness and knowledge, and there is strong support for notifying patients of breast density, as well as interest in education and training in the subject area. Breast density has gained increased attention from policy-makers, practitioners, and consumer advocates in recent years and will continue to attract discussions with increasing evidence surrounding it. As GPs play a central role in cancer prevention and control, their involvement in discussions related to density notification, evaluation and appraisal of evidence, development of communication strategies, and participation in ongoing research on the topic will be indispensable.

## Figures and Tables

**Table 1 ijerph-19-09029-t001:** General practitioner characteristics (N = 60). Data are presented as n (%).

Characteristic	No. of Participants (%)
**Public vs. Private**	
Public	7 (12)
Private	45 (75)
Both	8 (13)
**Type of practice**	
Solo practice	5 (8)
Group practice	48 (80)
Hospital clinic	5 (8)
Other	2 (3)
**Location of practice**	
Major city	40 (69)
Inner and outer regional	16 (27)
Remote and very remote	2 (4)
**State**	
Australian Capital Territory (ACT)	5 (8)
New South Wales (NSW)	20 (33)
Victoria (VIC)	11 (18)
Queensland (QLD)	7 (12)
South Australia (SA)	3 (5)
Western Australia (WA)	11 (18)
Tasmania (TAS)	2 (3)
Northern Territory (NT)	0 (0)
Missing	1 (2)
**Gender identity**	
Female (F)	54 (90)
Male (M)	6 (10)
**Years of experience in general practice**	
<10	30 (50)
10–19	13 (22)
20–29	8 (13)
30+	9 (15)
**Country of primary medical degree**	
Australia	49 (82)
Other	11 (18)
**Average clinical work hours/week**	
<10	4 (7)
10–19	8 (13)
20–29	19 (32)
30–39	21 (35)
40–49	8 (13)
**Number of patients managed/week**	
<50	29 (48)
50–99	22 (37)
100+	9 (15)
**Estimated proportion of female patients aged** 40+	
<25%	4 (7)
25–49%	32 (53)
50–74%	18 (30)
75–100%	6 (10)
**Specific interest in women’s or breast health**	
Yes	21 (35)
No (other or no specific interest reported)	39 (65)

**Table 2 ijerph-19-09029-t002:** Knowledge of breast density.

Items	No. of Participants (%)
**Prior to taking this survey, have you heard or read anything about the term ‘breast density’?**	
Yes	58 (97)
No	2 (3)
**How did you hear or read about it? (multiple choice)** ^**a**^	*n* = 58
Medical school education	12 (21)
Clinical work experiences	27 (47)
Reading mammogram reports	43 (74)
Attending professional and academic conferences	21 (36)
Attending talks or seminars	22 (38)
Reading journal articles or other academic/professional reading materials	27 (47)
Talking to colleagues or other clinicians	27 (47)
Personal experiences or experiences of close family or relatives	17 (29)
Other	1 (2)
**How well do you feel you understand breast density as a clinical issue at the moment? (range 1–5; 1—no understanding, 3—average level of understanding, 5—high level of understanding)**	Mean = 3.327; SD = 0.85
**What is your understanding about how breast density is measured? (multiple choice)**	*n* = 58
Physical examination of the breasts	13 (22)
Mammographic imaging	48 (83)
Ultrasound imaging	18 (31)
MRI imaging	18 (31)
Digital breast tomosynthesis (3D mammogram)	32 (55)
Other	0 (0)
Don’t know	4 (7)
**Have you heard of, or are you familiar with, the Breast Imaging Reporting and Data System** **(BIRADS) on classifying and reporting breast density?**	
Yes	40 (69)
No	18 (31)
**Having dense breasts can increase the risk of developing breast cancer.**	
True	36 (62)
False	12 (21)
Don’t know	10 (17)
**If a woman has dense breasts, what impact does this have on the ability of a mammogram to** **correctly detect cancer?**	
Dense breasts make it easier to see cancer on a mammogram	1 (2)
Dense breasts do not impact the ability to see cancer on a mammogram	3 (5)
Dense breasts make it more difficult to see cancer on a mammogram	53 (91)
Don’t know	1 (2)
**Does a woman’s breast density change with age?**	
No, it does not change with age	1 (2)
Usually increases with age	5 (9)
Usually decreases with age	48 (83)
Don’t know	4 (7)
**Prior to this survey, were you familiar with the concept of “supplemental screening” in the discussion of breast density? (i.e., offering patients with dense breasts a supplemental screening exam in addition to mammography in order to exclude breast cancer)**	
Yes	42 (72)
No	16 (28)
**Evidence suggests the following supplemental screening tests may be warranted for women with dense breasts (multiple choice):**	
Ultrasound	30 (52)
MRI	26 (45)
Tomosynthesis (3D mammography)	21 (36)
Genetic testing	3 (5)
None of the above	1 (2)
Don’t know	2 (3)

^a^ Only those who had prior knowledge or experience were analysed for the subsequent items.

**Table 3 ijerph-19-09029-t003:** Prior experiences and practice.

Items	No. of Participants (%)
**How often does a discussion about breast density typically occur in your own clinical practice?**	*n* = 58
Every day	7 (12)
About once a week	9 (16)
About once a month	23 (40)
About once a year	13 (22)
Never. I have never discussed breast density with a patient	6 (10)
**Discussion about breast density never came up because (multiple choice):** ^a^	*n* = 6
It was not reported on the mammogram reports	4 (67)
Patients did not ask about it	1 (17)
I do not have adequate knowledge about breast density and its implications	2 (33)
I have not come across the idea of breast density as an issue to discuss with patients	3 (50)
**How comfortable are you answering patients’ questions about breast density?** ^b^	*n* = 52
Very comfortable	16 (31)
Somewhat comfortable	26 (50)
Not comfortable	9 (17)
I am not asked	1 (2)
**Which best describes your approach about supplemental screening?**	
Every patient with dense breasts is offered a supplemental screening.	16 (31)
Only certain patients with dense breasts are offered supplemental screening, based on unique patient/risk factors.	14 (27)
Only some patients with dense breasts are offered supplemental screening, based on recommendation in the mammogram report.	14 (27)
I don’t offer supplemental screening based on breast density	8 (15)
Which supplemental screening modality do you most commonly suggest to your patients with dense breasts? ^c^	*n* = 44
Digital Breast Tomosynthesis (DBT)	7 (16)
Breast Ultrasound	26 (59)
Magnetic Resonance Imaging (MRI)	9 (20)
None	2 (5)
**When suggesting a particular supplemental imaging study, which one of the following most heavily influences your decision?**	
Ease of obtaining exam	10 (23)
Patient preference	3 (7)
Patient’s overall breast cancer risk	17 (39)
Costs to patient	8 (18)
I only have one exam to offer	2 (5)
Other (please specify)	4 (9)
**How well equipped do you think primary care clinicians in your state are to discuss breast density with women at this time?** ^d^	*n* = 56 Mean = 2.714, SD = 0.80
**What do you think are/would be the key challenges in discussing breast density with patients? Please tick all that apply.**	*n* = 58
Clinician’s own lack of knowledge	49 (84)
Patient’s low health literacy	24 (41)
Patient with a low income	20 (34)
Patient’s low educational background	10 (17)
Lack of communication resources	24 (41)
Lack of time	21 (36)
No challenges	1 (2)
Other	3 (5)

^a^ Only those who had no experience in discussing breast density were analysed for this item. ^b^ Only those who had experience in discussing breast density were analysed for the subsequent items. ^c^ Only those who recommend supplemental screening were analysed for the subsequent items. ^d^ Only those who had no experience in discussing breast density were analysed for this item.

**Table 4 ijerph-19-09029-t004:** Views towards notification.

Items	No. of Participants (%)
**Do you believe that Australia should make it mandatory for publicly funded screening services to report breast density if women have dense breasts?**	
Yes	45 (75)
No	3 (5)
Don’t know	12 (20)
**I believe notifying women of their breast density will promote informed decision-making.** ^a^	*n* = 58
Strongly disagree	3 (5)
Somewhat disagree	1 (2)
Neither agree nor agree	10 (17)
Agree	27 (47)
Strongly agree	17 (29)
**I am in favour of a policy to inform women of their breast density as part of their mammogram results.**	
Strongly disagree	2 (3)
Somewhat disagree	3 (5)
Neither agree nor agree	7 (12)
Agree	28 (48)
Strongly agree	18 (31)
**Counselling women about breast density is primarily my responsibility.**	
Strongly disagree	2 (3)
Somewhat disagree	5 (9)
Neither agree nor agree	11 (19)
Agree	30 (52)
Strongly agree	10 (17)
**Notifying women about their breast density may cause undue anxiety to them.**	
Strongly disagree	4 (7)
Somewhat disagree	10 (17)
Neither agree nor agree	14 (24)
Agree	26 (45)
Strongly agree	4 (7)
**I believe women have the right to know about their breast density**	
Strongly disagree	2 (3)
Somewhat disagree	1 (2)
Neither agree nor agree	2 (3)
Agree	34 (59)
Strongly agree	19 (33)
**If Australia were to adopt routine and widespread notification of breast density after each mammogram, how would this policy affect your clinical practice?**	
**This could affect my clinical practice.**	
Strongly disagree	4 (7)
Somewhat disagree	4 (7)
Neither agree nor agree	10 (17)
Agree	34 (59)
Strongly agree	6 (10)
**I would feel prepared to respond to requests from patients who are notified that they have dense breasts.**	
Strongly disagree	4 (7)
Somewhat disagree	6 (10)
Neither agree nor agree	9 (16)
Agree	30 (52)
Strongly agree	9 (16)
**I would need training in how to respond to requests from patients who are notified that they have dense breasts.**	
Strongly disagree	5 (9)
Somewhat disagree	7 (12)
Neither agree nor agree	11 (19)
Agree	25 (43)
Strongly agree	10 (17)
**The breast density notification mandate may include several specific components. Which of the following statements do you think should be included in the letters provided to women? (multiple choice)**	
The mammogram showed dense breast tissue	27 (47)
The mammogram showed this particular degree or category of density	35 (60)
Dense breast tissue increases the risk of breast cancer	23 (40)
Dense breast tissue is common	37 (64)
Dense breast tissue can make it more difficult to find cancer on a mammogram	37 (64)
Additional screening may be advisable	44 (76)
Pros and cons of additional screening	28 (48)
MRI or ultrasound is the best means to find potential cancers in dense breasts	18 (31)
The patient should discuss results with their primary care physician or the referring physician	48 (83)
The patient has a right to discuss results with a radiologist	12 (21)

^a^ Not all participants responded.

## Data Availability

Data are available upon reasonable request.

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
