# Peer review of "Australian General Practitioners’ Current Knowledge, Understanding, and Feelings Regarding Breast Density Information and Notification: A Cross-Sectional Study"

_ijerph, 2022, doi:10.3390/ijerph19159029_

Round 1

Reviewer 1 Report

Dolan and collaborators present a descriptive article. They made a questionary to several doctors regarding breast density information. 

They recognize that they have a small size sample. However, analyze and describe quite well the presented information.

Authors should check carefully the bibliography. For example: 

Qualtrics, 2021. Qualtrics. Qualtrics, Provo, Utah, USA.

Author Response

We thank the Reviewer #1 for reviewing our manuscript and for the positive feedback. As suggested, we have checked the bibliography and now changed the reference to the Qualtrics software to the following: Qualtrics, 2021. Qualtrics. Provo, Utah, USA. Available at https://www.qualtrics.com

Reviewer 2 Report

Dolan et al. conducted a cross-sectional survey to collect quantitative data from Australian GPs on breast density measurements. Most GPs knew about breast density and nearly 90% discussed breast density with patients. Three quarters supported making breast density notification mandatory for patients with dense tissue. Most wanted more education on breast density. The authors concluded GPs strongly support notifying patients of breast density, and showed interest in further education and training. A limitation was the small sample size, likely related to COVID.

This reviewer has the following comments/questions:

1. A limitation of the study is the low participation rate in the questionnaire by GPs. Is there any information about how the GPs whio declined differed from those that accepted? Did they have  allowed interest or knowledge about breast density?

2. Line 198: Leave out "a". Should Nickel et al. be included at the start of the sentence?

Author Response

We thank the reviewer #2 for the careful examination of our manuscript. As pointed out, the small sample size is one of the major limitations of our study and this has been addressed in the Discussion section.

This study is an online questionnaire study and the first question the potential participants were asked was to indicate their consent to take part in the survey. If participants chose “no” in their answer, they survey was terminated for them. Therefore, we did not collect any demographic and clinical characteristics information for those GPs who declined to participate; and will not be able to compare their characteristics with those who proceeded with the questionnaire.  

Those GPs who proceeded with the questionnaire were asked about their knowledge about breast density and a detailed analysis of their responses is provided in Table 2. Most of the GPs (n=58, 97%) had heard or read about breast density prior to taking the survey. Please see pages 6-7.

We have now also revised this sentence suggested by the reviewer in point #2. We have kept “a” and changed “a studies” to “a study”. We have also added Nickel et al at the start of the sentence. Please see page 10.

Reviewer 3 Report

In the manuscript titled “Australian General Practitioners’ current knowledge, understanding and feelings regarding breast density information and notification: a cross-sectional study” by Dolan et al, the authors have used a small cohort of participating GPs to assess their knowledge about breast density information and their views towards notifying patients of the same.

The manuscript is well written, results are presented in a clear manner and the methods are adequately described. Although the manuscript does suffer from the drawback of small sample size, the authors have addressed this in the discussion section and have not made any bold claims about the applicability of this data to a broader population, so the data can be accepted at its face value.

Minor concerns:

The data presented in Table 1 has different formatting compared to tables 2-4. Please highlight the sub-headings in Table 1 (Type of practice, state, etc.) by changing their font to bold.

Table 2: Under the first heading, the last column has an unnecessary “%” (“3%”).

The data in Table 1-4 is all presented in “parts of whole” format (percentages). As such, I would suggest a graphical representation of this data. Instead of the tabular presentation, pie-charts would perhaps help the readers get a good idea of the data in one glance and it would be more visually pleasing too. This is just a suggestion though, and the authors do not need to comply if they deem tabular presentation more fit.

I recommend that the manuscript can be accepted in its present format after correcting the two minor typographical/formatting errors I mentioned above.

Author Response

We thank the reviewer #3 for the thorough reading of our manuscript and for the positive and valuable feedback.

Minor concerns:

The data presented in Table 1 has different formatting compared to tables 2-4. Please highlight the sub-headings in Table 1 (Type of practice, state, etc.) by changing their font to bold.

As suggested, the subheading formatting in the Table 1 has been changed to bold. Please see pages 5-6.

 Table 2: Under the first heading, the last column has an unnecessary “%” (“3%”).

We have now removed % from “3%”. Please see page 6.

 The data in Table 1-4 is all presented in “parts of whole” format (percentages). As such, I would suggest a graphical representation of this data. Instead of the tabular presentation, pie-charts would perhaps help the readers get a good idea of the data in one glance and it would be more visually pleasing too. This is just a suggestion though, and the authors do not need to comply if they deem tabular presentation more fit.

We thank the reviewer for this suggestion, and we agree with the reviewer that making visual representation of the data using pie-charts would greatly enhance presentation and accessibility. However, given the volume of outcomes (around 40) across all tables to be presented, we would have to create a large number of pie-charts to be able to cover them all. Therefore, we feel that tabular presentation is a better fit in this instance.

I recommend that the manuscript can be accepted in its present format after correcting the two minor typographical/formatting errors I mentioned above.

 We are grateful for the reviewer for recommending acceptance of our manuscript.